# Study of Farmers' Willingness to Participate in Environmental Governance Based on Recycling, Reduction and Resourcing

**Jia Liu [1], Senwei Huang [2] and Yijia Wang [1,*]**

[1] School of Urban Economics and Public Administration, Capital University of Economics and Business, Beijing 100070, China; liujia@cueb.edu.cn
[2] School of Public Administration & Law, Fujian Agriculture and Forestry University, Fuzhou 350002, China; hsw@fafu.edu.cn
[*] Correspondence: wyj2609596@163.com

**Abstract:** In order to establish a green and low-carbon agricultural economic system based on the principles of recycling, reduction and resource utilization, to promote a comprehensive green transformation of the economic development and to achieve carbon peaking and carbon neutrality, this article examines the farmers' willingness to participate in rural environmental governance. Through a questionnaire survey in the Fujian, Anhui and Shanxi provinces, this article explores the influence of homogeneous and heterogeneous relationships in social networks on the farmers' willingness to participate in rural environmental governance using a logit model and tries to reveal the deeper mechanisms. The results show that: (1) heterogeneous relationships have a significant positive effect on farmers' participation in rural environmental governance, but homogeneous relationships do not have a significant effect. (2) The larger the size of the social network, the weaker the farmers' willingness to participate in rural environmental governance. (3) Age and education level have significant effects on willingness to participate. The older the age, the weaker the willingness to participate; the higher the education level, the stronger the willingness to participate. (4) The larger the number of family members, the stronger the farmers' willingness to participate in environmental governance. (5) The subjective cognitive status of farmers also has an important influence on their willingness to participate. The more environmental knowledge is acquired, the stronger the willingness to participate in rural environmental governance. Therefore, to promote rural environmental management, there is an urgent need to keep modern farmers on rural land and to make the countryside a beautiful space for living and working with complete living functions.

**Keywords:** recycling; reduction and resourcing principles; rural environmental governance; willingness to participate

## 1. Introduction

The report of the twentieth National Congress of the Communist Party of China states that Chinese modernization is a modernization in which people and nature live together in harmony. Nature is the basic condition for human beings to survive and develop. Respect for nature, being in line with nature, and protecting nature are the comprehensive construction of a socialist modernization of the country's inherent requirements. It is necessary to firmly establish and practice the concept of green water and green mountains as the silver mountain of gold, standing at the height of the harmonious coexistence of man and nature to plan development [1].

To accelerate the green transformation of development and construct China into a beautiful country, we must focus on solving outstanding environmental problems, especially the rural environment. With the acceleration of "industrialization" and "urbanization", rural areas, as supporters of urban ecosystems, have been the abatement side of urban pollution. The ecological environment in rural areas has paid a tragic price. As agricultural

surface source pollution is influenced by many factors such as the development concept of "pursuing growth", "urban-rural dualistic economic and social structure", high cost of treatment, the diversification of farmers' production behavior, the land system, related policies, market factors, farmers' perceptions and changes in a social environment [2], the problem of rural environmental management has become more and more complex [3], which seriously affects the sustainable development of the rural society and economy.

The deterioration of the rural ecological environment is the biggest obstacle to the establishment of ecological agriculture and the construction of a new countryside in China. Therefore, it is necessary to strengthen the prevention and control of pollution from agricultural sources, carry out rural habitat improvement actions and build a diversified environmental governance system with common public participation. The public, mainly farmers, as the direct bearers of rural environmental damage, are also the direct beneficiaries of rural environmental management. Farmers are both the direct bearers of rural environmental damage and the direct beneficiaries of rural environmental management. This means that in rural areas, bringing farmers into rural environmental management is an effective guarantee for the success of all efforts [4]. The farmers' participation in rural environmental management is not optimistic due to a series of reasons, such as their lack of environmental awareness. To effectively improve the rural environment, it is necessary to analyze in depth what factors influence the farmers' participation in rural environmental management. Studies have shown the role of social capital accumulation in reconciling the contradictions between economic growth and environmental protection [5]. In the current context of rapid rural transformation, rural social capital has become an indispensable component of environmental governance, even directly affecting its effectiveness. There are both positive and possible negative impacts of social capital on rural environmental governance. This depends on the adequacy of the stock of social capital possessed by the village [6]. However, as an important part of social capital, social networks have rarely been studied separately. Whether social networks have a significant impact on the farmers' willingness to participate in environmental governance is the central issue of our study.

## 2. Review of the Relevant Literature and Theoretical Foundation

### 2.1. Review of the Relevant Literature

"Replacement, Reduction, and Refinement", known as the "3R" principle, was first proposed in the book "Principles of Compassionate Experimental Techniques" (1959), co-authored by zoologist Russell and microbiologist Burch. The international scientific community has widely recognized the "3R" principle, and all countries are actively implementing the "3R" principle. Under the concept of circular economy development, the "3R" principle can be interpreted as "Recycling, Reduction, and Resourcing".

Fei Xiaotong (2012), in his book "From The Soil", refers to the grassroots structure of Chinese vernacular society as a "differential pattern", that is, a social network made up of individual private connections" [7]. In this "differential pattern", the morals and norms in each person's heart vary in the degree to which they are imposed, depending on the proximity of the other person to the "self". The important role of social networks was confirmed by studies. Pierre (1992) believes that social networks and social relationships facilitate the advancement of actors in different fields [8]. Rogers (2010) has suggested that social networks have an important influence on the diffusion of agricultural technologies and on the attitudes and behaviors of farmers toward their adoption [9]. Skaalsveen (2020) also suggests that interpersonal networks are important to farmers and influence their learning and decision making, and that farmers tend to view farmers with high levels of knowledge and experience as their primary source of information [10]. Dadzie (2022) has suggested that the effectiveness and usefulness of social interactions and a high level of trust in social networks for cassava growers can reduce their risk-averse behavior [11]. Many domestic scholars have also confirmed the role of social networks in influencing farmers' willingness to participate in ecological compensation [12], farm switching out behavior [13], willingness to participate in irrigation management reform [14], planting behavior [15], and

green prevention and control technology adoption behavior through empirical studies [16]. In terms of access to agricultural information channels, farmers trusted their neighbors the most [17]. Jiang (2021) found that the interaction of media channels and social interactions promoted the farmers' adoption of straw return and mutually reinforced each other [18]. In traditional Chinese culture focusing on interpersonal relationships, social networks have become an essential capital for information dissemination and resource exchange. Granovetter (1983) proposed the theory of heterogeneous relationships, which divides social relationships in social networks into "homogeneous relationships" and "heterogeneous relationships" [19]. He believes that homogeneous relationships are those with more interactions, deeper feelings, closer relationships or more reciprocal exchanges. The opposite is true for heterogeneous relationships. Homogeneous relationships are connections that occur within groups, and the dissemination of information is repetitive. In contrast, heterogeneous relationships are connections between different circles of interaction, and the dissemination of information is differential [19]. Lin et al., 1981, have suggested that heterogeneous relationships are based on instrumental actions and consist mostly of weak relationships, while homogeneous relationships are based on affective actions and consist mostly of strong relationships [20]. Both they and Granovetter believe that heterogeneous relationships better facilitate the flow of information. In contrast, Yanjie Bian (2010) argues that in a social environment such as China, homogeneous relationships can also serve as a bridge to disseminate information [21]. In a cultural context that emphasizes relationships, interpersonal relationships are a key mechanism of social behavior in China [21]. Munsh (2004) also believes that homogeneous relationships are more conducive to information flow [22]. He pointed out that similar individual characteristics in homogeneous relationships are easier for information acquisition and dissemination [22].

Regarding the measurement of homogeneous and heterogeneous relationships, there are now more uniform indicators in academia. Granovetter proposes that homogeneous relationships are people who have reliable and stable relationships, strong emotional ties and more intimate relationships, for example, friends and family, neighbors and acquaintances, etc. Heterogeneous relationships are people who come from different circles and do not have many emotional ties, for example, work relationships or general acquaintances. It is characterized by a high degree of heterogeneity and a wide range and continuity. In their study, Mcpherson et al., 2001, included neighbors, hometowns and blood relatives with geo-relations in homogeneous relationships, while classmates, colleagues and fellow associations from social organizations were included in heterogeneous relationships [23]. Yanjie Bian has used the dimensions of "affinity, trust, and familiarity" to describe homogeneous relationships [21]. In their study, Yubei Li et al., 2018, have also classified family members, close friends and neighbors as homogeneous relationships and research extension agencies, NGOs, financial institutions, seed suppliers and others without intimate emotional maintenance as heterogeneous relationships [24,25].

In summary, however, the above literature mostly includes studies on social networks in social capital alongside trust and norms and contains few studies that analyze social networks separately empirically. At the same time, the research specifically focused on the farmers' participation in environmental governance needs to be expanded. In rural societies that focus on "blood", "kinship" and "local ties", relational networks and favors play an important role in the farmers' behavior and intentions. Therefore, what is the role of homogeneous and heterogeneous relationships in social networks in the farmers' willingness to participate in environmental governance decisions is our main research question.

### 2.2. Theoretical Foundation

From Popkin's rational small farmer perspective, the economic level of farmers affects their environmental behavior from an input–output perspective. The farmers' low income levels and weak economic capacity to pay for the costs of environmental behavior lead farmers not to participate in environmental governance. In addition, "farmers' attitudes

toward environmental responsibility vary according to farm size (both physical and economic)". Large farms are more focused on eco-efficiency and cleaner production. A central concern of economic factors is the relationship between the profitability of individual farmers and farm households and the costs of environmental behavior. Factors such as household economic resources, loan support and access to relevant subsidies influence farmers' willingness to participate in the development of a new low-carbon countryside. Economic conditions play a fundamental role in social development, and imbalances in economic and social development also act on factors such as education, cognition and skills, resulting in an uneven regional distribution of the farmers' participation levels. In regions with a developed economy, dense population, sound transportation infrastructure construction or developed eco-tourism and poverty alleviation tourism, the labor market is broad; the farmers have more exposure to the outside environment and access to information, which has an educational effect on environmental awareness; and the farmers' willingness to participate in environmental management may be greater [26–28]. On the contrary, the farmers' willingness to participate may be smaller. In addition, the later emergence of pro-poor tourism can bring environmental benefits in addition to economic, social and cultural benefits. However, at the same time, excessive tourism may also lead to the degradation of the farmers' quality of life, such as soil pollution due to heavy metals emitted from transportation industrial dwellings and domestic waste overload from food consumption, making farmers more dependent on government departments, exhibit less proactive environmental behavior, and show less willingness to participate in environmental governance [29].

The system as a specific code of conduct regulates the space and form of the participation of the farmers. In environmental governance, "a sound legal system for environmental protection is an important prerequisite for realizing citizens' right to environmental participation". The lack of a systemic legal system makes the farmers vulnerable to environmental violations, and the lack of administrative safeguards and judicial remedies as an extension of the system also leads to violations of the farmers' environmental rights, making the farmers' ecological participation limited. The system also generates trust capital, and institutional trust is a robust promoter of the farmers' participation in environmental governance.

The scale of the relational reciprocity network formed by socialized small farmers in the acquaintance society is small, so the insufficient stock of social capital can affect the effectiveness of environmental governance. Social capital has a value-oriented function in enabling environmental behavior, and cognitive social capital can shape the farmers' pro-environmental values through value internalization and value sharing. Conversely, otherness and antagonistic tolerance can make social conflicts deepen or even move toward division [30].

## 3. Data Source, Variable Selection and Model Setting

### 3.1. Data Sources

The data used are from the field research data of the Chinese National Social Science Project team. From July 2017 to July 2018, members of the group investigated 102 villages in 10 localities, including Fujian Province, Anhui Province and Shanxi Province. The research method is a field survey. The survey included basic information about the farmers' households, their domestic waste disposal, their subjective perceptions, their willingness to protect the environment and their social capital in rural areas. A total of 529 questionnaires were obtained. According to the purpose of this paper, after eliminating invalid questionnaires, 343 valid questionnaires were obtained. From the distribution of the sample, the age of the survey respondents was mainly concentrated at the age of 50 or above, accounting for 66.16%. Among them, the highest proportion was over 60 years old, and the least number of people were under 20 years old. It can be seen that the current rural elderly are predominant. During the survey, we learned that most of the young and middle-aged people in the rural areas went to work abroad. The village was mainly inhabited by the

elderly and children left behind. Of the survey respondents, the highest percentage had primary and secondary education. The proportion who had a college education and above was the least, and the proportion who were illiterate was 17.58%. This indicates that the current education level of the rural population is generally concentrated in primary and junior high school, accounting for 70.7%. Only 4.92% of the population has received higher education. The physical health condition of the survey respondents was concentrated in the relatively good range. The percentage of those with poor and very poor physical health was relatively small. The proportion of party members only accounted for 9.71, and the proportion of the masses accounted for 90.29%, which shows that the proportion of party members in rural areas is smaller.

### 3.2. Variable Selection and Descriptive Statistics

Referring to the existing literature, the farmers' willingness to participate in rural environmental governance was chosen as the dependent variable in this paper. A total of 16 variables, including five categories of individual characteristics, family characteristics, village characteristics, subjective cognitive situations and social networks, were selected as independent variables (see Table 1).

**Table 1.** Basic characteristics of the sample.

| Projects | | Frequency | Frequency |
|---|---|---|---|
| Age | Age ≤ 20 | 3 | 0.57% |
| | 20 < Age ≤ 30 | 30 | 5.67% |
| | 30 < Age ≤ 40 | 45 | 8.51% |
| | 40 < Age ≤ 50 | 101 | 19.09% |
| | 50 < Age ≤ 60 | 144 | 27.22% |
| | Age > 60 | 206 | 38.94% |
| Health Status | Very poor | 7 | 1.33% |
| | Poor comparison | 70 | 13.28% |
| | General | 101 | 19.17% |
| | Comparatively good | 240 | 45.54% |
| | Very good | 109 | 20.68% |
| Education level | Illiterate | 93 | 17.58% |
| | Primary School | 189 | 35.73% |
| | Junior High School | 185 | 34.97% |
| | High School | 36 | 6.81% |
| | College | 13 | 2.46% |
| | College or above | 13 | 2.46% |

Willingness to Participate. The dependent variable of this model is willingness to participate. Specifically, it is measured by the question: "Are you willing to participate in rural environmental management?". The article converts the five categories of willingness into two categories, as shown in Table 1. The mean value of the farmers' willingness to participate is 0.676; this indicates that more than half of the farmers are willing to participate in rural environmental management.

Social network. (1) Social network size. Referring to the existing literature, we chose "the number of contacts in a cell phone or WeChat address book" to characterize the "social network size". Li Yubei has selected the monthly cell phone bill to measure the size of the social network [24,25], and Acharya, A et al., 2017, have used the expenditure on human gifts to measure the size of social network, but both of these measures are lacking [26]. First of all, with today's fast-changing Internet, most people no longer rely on phone calls to maintain contact with friends but contact each other through mobile communication software. During the research process, we found that even in rural areas, the use of WeChat has long been popular, and most people said that the use of WeChat is very convenient and they do not have to pay phone bills, and most of the contact with relatives and friends is performed using WeChat. Second, the expenditure on favors and courtesies can vary

greatly depending on the region and customs. The economic development of the eastern coastal region is more developed, and the expenditure on human courtesy is larger, while the economic development of the western region is more backward, and the expenditure on human courtesy is relatively small, so it is not reasonable to use the expenditure on human courtesy to measure the scale of the social network. Therefore, based on the current background of cell phone network popularity and the regional differences in our research sites, we use "the number of cell phone or WeChat contacts" to measure the "social network size". (2) Homogeneous relationships. Homogeneous relationships are characterized by "closeness to relatives", "closeness to neighbors" and "closeness to acquaintances". This is measured by the "frequency of meeting and gathering among relatives", "frequency of chatting and gathering among neighbors", and "frequent greeting of acquaintances". The frequencies include "never, several times a year, several times a month, several times a week, and every day", with 1–5 representing each of the above five options. The options for "greeting acquaintances often" included "strongly disagree, somewhat disagree, usually, somewhat agree, and strongly agree", with values from 1 to 5. The above questions indicate the degree of their homogeneous relationships, respectively. As shown in Table 2, the mean values of the three homogeneous relationship variables were above 2.5, indicating that the homogeneous relationships of farmers were more developed. (3) Heterogeneous relationships. Heterogeneous relationships were characterized by three indicators: "closeness to village officials", "closeness to government personnel" and "closeness to social groups". Specifically, the questions were measured by "frequent participation in village cadres' election meetings", "frequent contact with government personnel" and "frequent participation in parties, religious activities, volunteer groups, mutual aid associations, etc.". The first two questions contained five options: "strongly disagree, relatively disagree, generally agree, relatively agree, and strongly agree", which were assigned values of 1–5, respectively, representing the degree of heterogeneity of their relationships. The last question contained two options, "yes" and "no", and was assigned the values of 1 and 0. As seen in Table 2, only the mean value of "closeness to village officials" among the heterogeneous relationships is greater than 2.5. Among the heterogeneous relationships, only the mean value of "closeness to village cadres" is greater than 2.5, while the other two variables are below the mean value. It can be seen that the farmers' heterogeneous relationships are less developed.

**Table 2.** Definition of variables and descriptive statistical analysis.

| Variable Name | Variable Name | Meaning and Assignment | Average Value | Standard Error |
|---|---|---|---|---|
| Dependent Variable | Willingness to Participate | 1 = willing; 0 = unwilling | 0.676 | 0.469 |
| | Individual Characteristics | | | |
| | Age | Contact Variables | 55.509 | 13.626 |
| | Education level | Continuous Variables | 6.143 | 3.943 |
| | Health Status | 1 = very poor, 2 = poor, 3 = fair, 4 = better, 5-very good | 3.710 | 0.983 |
| | Family Characteristics | | | |
| IndpendentVariables | Number of Family Members | Continuous Variables | 5.013 | 2.224 |
| | Annual household income | Continuous Variables (10,000 Yuan) | 6.487 | 6.794 |
| | Village Features | | | |
| | Is a Project Village | Whether the village is a rural environment continuous improvement project 1 = yes, 0 = no | 0.513 | 0.500 |

**Table 2.** *Cont.*

| Variable Name | Variable Name | Meaning and Assignment | Average Value | Standard Error |
|---|---|---|---|---|
| | Distance from Town | Continuous Variables | 6.977 | 9.337 |
| | Subjective Cognitive Status | Continuous Variables | 30.081 | 3.938 |
| | Size of Social Network | Number of cell phone or WeChat address book Continuous Variables | 82.105 | 132.441 |
| | Homogeneous Relatioshihips | | | |
| | Intimacy with Relatives | Frequency of meetings and gatherings between relatives 1 = Never, 2 = Several times a year, 3 = Several times a month, 4 = Several times a week, 5 = Every day | 3.138 | 1.247 |
| | Intimacy with Neighbors | Frequency of chats and gatherings between neighbors 1 = Never, 2 = Several times a year, 3 = Several times a month, 4 = Several times a week, 5 = Every day | 4.045 | 1.207 |
| | Intimacy with Acquaintances | Greet acquaintances often 1 = Strongly disagree, 2 = Rather disagree, 3 = Generally, 4 = Rather agree, 5 = Strongly agree | 4.331 | 1.046 |
| | Heterogeneous Relationships | | | |
| | Intimacy with Village Officials | Regularly participate in village officer election meetings 1 = Strongly disagree, 2 = Rather disagree, 3 = Generally, 4 = Rather agree, 5 = Strongly agree | 3.727 | 1.429 |
| | Intimacy with Government Personnel | Frequent contact with government personnel 1 = Strongly disagree, 2 = Rather disagree, 3 = Generally, 4 = Rather agree, 5 = Strongly agree | 2.421 | 1.381 |
| | Intimacy with Social Groups | Whether they regularly participate in parties, religious activities, volunteer groups, mutual aid associations, etc. 1 = yes, 0 = no | 0.378 | 0.485 |

Control variables. In terms of the selection of control variables, individual characteristics, family characteristics, and village characteristics are influential factors that may affect willingness to participate in environmental governance. A large number of empirical studies have proven that the willingness to participate is closely related to the individual characteristics and family characteristics of the respondents. Therefore, in this paper, "age, education level, health status" were selected as the dependent variables of individual characteristics concerning existing studies [27–29]. The number of family members and annual household income were selected as household characteristics. The village characteristics were selected as "whether it is a village in the rural environment continuous improvement project, distance from town". Table 2 shows that the average age of the respondents was

55.509, the average education level was 6.143 years, and the average health status was 3.725, i.e., between average and relatively healthy.

Subjective cognitive status. In addition, subjective cognitive status is also an important influencing factor. Behavioral intention is the most direct factor influencing actual behavior, and behavioral intention is influenced by behavioral attitude, subjective norms, and perceived behavioral control. Therefore, this paper selects four aspects to characterize environmental rights perception, personal efficacy perception, risk perception and responsibility perception, which are specifically measured by some 9 questions. Environmental rights perceptions specifically include the following 2 issues: "I have the right to know how funds for environmental protection projects are used, and I have the right to voice my opinion if I disagree with the village's environmental policy". The perception of personal effectiveness includes: "It is enough for the village committee to manage public affairs in the village, and it is not useful for individuals to take actions to protect the environment". Risk perceptions include: "At present, the quality of the rural environment is very serious, and environmental pollution has had a serious impact on my life". Responsibility perceptions include: "Everyone should protect the environment, the government and village committee should be the main body for environmental pollution management, and environmental protection is important for the development of rural areas". The options for the above nine questions included strongly disagree, relatively disagree, generally, relatively agree and strongly agree and were assigned values of 1–5 in that order. Finally, the overall subjective cognitive level was measured by summation, and the final scores ranged from 9 to 45, representing different levels of the subjective cognitive level, respectively. Table 2 shows that the average score of the subjective cognitive status of the questionnaire respondents is 30.081, which is greater than the mean value of 27, indicating that the current subjective cognitive status of the environment of farmers is better.

### *3.3. Model Setting*

Logistic regression models are nonlinear categorical statistical methods designed explicitly for the regression analysis of dichotomous dependent variables and to analyze the degree of influence of different factors on the dependent variable through regression modeling. Therefore, the paper uses a logistic stepwise regression model to estimate the regression parameters using the maximum likelihood estimation method. The regression parameters were estimated using maximum likelihood estimation. According to the requirements of the logistic regression model, suppose $X_1, \ldots, X_i$ is a set of vectors associated with $Y$. Let $P$ be the probability of the occurrence of the willingness to participate, and take the ratio $P/(1-P)$ logarithmically to obtain $\ln[P/(1-P)]$, which is the logistic transformation of $P$, labeled as $logit(P)$:

$$Y = \ln\left(\frac{P}{1-P}\right) = a + B_1 X_1 + B_2 X_2 + \ldots B_i X_i \tag{1}$$

$$P = \frac{\exp(a + B_1 X_1 + B_2 X_2 + \ldots B_i X_i)}{1 + \exp(a + B_1 X_1 + B_2 X_2 + \ldots B_i X_i)}$$

$P$ in Equation (1) denotes a variable with a dichotomous nature. In order to clearly and concisely estimate the influence of social networks on the probability of the occurrence of the farmers' willingness to participate in environmental governance $P$, the willingness to participate is simplified to a 0–1 type dependent variable, where the variable is 1, representing the farmers' willingness to participate, and the variable is 0, representing the farmers' unwillingness to participate. a is a constant term, i.e., the natural logarithm of the ratio when the independent variable takes all values of 0. $X_i$ denotes the $i$th factor affecting farmers' willingness to participate. $B_i$ is the partial regression coefficient of the logistic regression, which indicates the degree of influence of variable $X_i$ on $Y$ or $logit(P)$.

#### 4. Empirical Test

*4.1. Multiple Linear Regression*

Before the regression analysis, to prevent internal correlation between the variables of the six dimensions of homogeneous and heterogeneous relationships. We conducted a multiple cointegration test for each variable. In general, there is a degree of multicollinearity between variables when VIF > 3. The results of the test are shown in Table 3, and the degree of co-linearity between the respective variables is within a reasonable range.

**Table 3.** Multicollinearity test.

| Variables | VIF | 1/VIF |
|---|---|---|
| Age | 1.52 | 0.657 |
| Education Level | 1.31 | 0.765 |
| Health Status | 1.22 | 0.818 |
| Number of Family Members | 1.07 | 0.933 |
| Annual Household Income | 1.38 | 0.727 |
| Is the Project Village | 1.05 | 0.955 |
| Distance from Town | 1.04 | 0.964 |
| Subjective Cognitive Status | 1.07 | 0.938 |
| Social Network Scale | 1.39 | 0.720 |
| Intimacy with Relatives | 1.08 | 0.922 |
| Intimacy with Neighbors | 1.09 | 0.921 |
| Intimacy with Acquaintances | 1.14 | 0.879 |
| Intimacy with Village Officials | 1.23 | 0.816 |
| Intimacy with Government Personnel | 1.28 | 0.781 |
| Intimacy with Social Groups | 1.11 | 0.901 |
| Average VIF | 1.20 | |

VIF means variance inflation factor.

*4.2. Binary Logistic Regression*

The article used Stata 13.0 to perform logistic stepwise regression on 346 sample data, and in order to test the regression stability, the article was tested by replacing the Probit model. Model 1 is the baseline model, and model 2 is the Oprobit model. See Table 4.

**Table 4.** Regression results.

| Variable Name | Variable Name | Model 1 Coefficients | Model 2 Coefficients |
|---|---|---|---|
| Individual Characteristics | Age | −0.020 | −0.010 * |
| | | (0.013) | (0.005) |
| | Education Level | 0.090 ** | 0.016 |
| | | (0.038) | (0.017) |
| | Health Status | 0.156 | 0.110 * |
| | | (0.148) | (0.064) |
| Family Characteristics | Number of Family Members | 0.149 ** | 0.070 ** |
| | | (0.072) | (0.028) |
| | Annual HouseholdIncome | 0.017 | (0.005) |
| | | (0.026) | (0.011) |
| Village Features | Is a Project Village | (0.391) | 0.010 |
| | | (0.278) | (0.121) |
| | Distance from Town | 0.000 | (0.006) |
| | | (0.015) | (0.007) |

**Table 4.** *Cont.*

| Variable Name | Variable Name | Model 1 Coefficients | Model 2 Coefficients |
|---|---|---|---|
| Subjective Cognitive Status | Subjective Cognitive Status | 0.106 ** | 0.021 |
| | | (0.042) | (0.016) |
| Size of Social Network | Size of Social Network | 0.000 | 0.001 |
| | | (0.001) | 0.000 |
| Homogeneous Relationships | Intimacy with Relatives | 0.108 | 0.077 |
| | | (0.117) | (0.051) |
| | Intimacy with Neighbors | 0.108 | 0.107 ** |
| | | (0.117) | (0.052) |
| | Intimacy with Acquaintances | 0.040 | 0.036 |
| | | (0.145) | (0.060) |
| Heterogeneous Relationships | Intimacy with Village Officials | 0.270 ** | 0.144 *** |
| | | (0.106) | (0.047) |
| | Intimacy with Government Personnel | 0.319 ** | 0.106 ** |
| | | (0.125) | (0.049) |
| | Intimacy with Social Groups | 1.123 *** | 0.535 *** |
| | | (0.413) | (0.165) |
| Constant Term | | −6.075 *** | / |
| | | (1.847) | / |
| Regional Variables | | Control | Control |
| Number of Samples | | 346.000 | 346.000 |
| Pseudo R2 | | 0.194 | 0.097 |

Note: ***, ** and * indicate 1%, 5% and 10% significance levels, respectively. Standard errors are in parentheses. All figures are rounded.

*4.3. Subsample Regression*

To further test the robustness of the regression results, the article divided the samples into high and low groups according to the median levels of age, education level, health status, number of household members, household income status and gender, respectively. See Table 5.

**Table 5.** Subsample regression results.

| Variable Name | Variable Name | The Older | The Younger | Higher Level of Education | Lower Education Level | Health Status Is Better | Poor Health | More Family Members | Fewer Family Members | Higher Household Income | Lower Household Income | Man | Woman |
|---|---|---|---|---|---|---|---|---|---|---|---|---|---|
| Individual Characteristics | Age | (0.031) | (0.017) | (0.005) | −0.061 ** | (0.025) | (0.012) | (0.005) | −0.037 * | (0.001) | (0.034) | (0.021) | (0.018) |
| | | (0.035) | (0.025) | (0.015) | (0.026) | (0.016) | (0.026) | (0.018) | (0.020) | (0.019) | (0.022) | (0.017) | (0.021) |
| | Education Level | 0.145 ** | 0.065 | 0.113 | 0.107 | 0.118 ** | 0.080 | 0.139 ** | 0.077 | 0.113 * | 0.096 * | 0.096 * | 0.106 * |
| | | (0.059) | (0.060) | (0.082) | (0.107) | (0.048) | (0.079) | (0.058) | (0.058) | (0.059) | (0.057) | (0.058) | (0.059) |
| | Health Status | (0.006) | 0.237 | 0.327 | (0.207) | 0.341 | (0.400) | 0.319 * | 0.415 | 0.105 | 0.042 | (0.135) | 0.313 |
| | | (0.212) | (0.225) | (0.200) | (0.234) | (0.416) | (0.468) | (0.191) | (0.270) | (0.217) | (0.212) | (0.219) | (0.226) |
| Family Characteristics | Number of Family Members | 0.084 | 0.174 | 0.138 | 0.211 * | 0.215 ** | 0.037 | 0.293 * | 0.104 | 0.164 | 0.155 | 0.228 ** | 0.083 |
| | | (0.090) | (0.131) | (0.097) | (0.123) | (0.098) | (0.121) | (0.151) | (0.279) | (0.125) | (0.096) | (0.111) | (0.107) |
| | Annual House-holdIncome | 0.106 * | (0.012) | 0.005 | 0.038 | 0.012 | 0.028 | 0.006 | 0.013 | 0.032 | 0.193 | 0.016 | 0.005 |
| | | (0.062) | (0.030) | (0.038) | (0.045) | (0.028) | (0.073) | (0.039) | (0.045) | (0.034) | (0.176) | (0.043) | (0.039) |
| Village Features | Is a Project Village | (0.661) | (0.173) | (0.179) | −0.877 * | (0.177) | (0.728) | (0.401) | (0.332) | (0.632) | (0.208) | (0.130) | (0.476) |
| | | (0.416) | (0.406) | (0.376) | (0.451) | (0.363) | (0.497) | (0.385) | (0.453) | (0.411) | (0.414) | (0.430) | (0.415) |
| | Distance from Town | (0.021) | 0.047 | 0.002 | (0.014) | (0.007) | (0.003) | (0.014) | 0.006 | 0.008 | (0.023) | (0.015) | 0.004 |
| | | (0.020) | (0.037) | (0.019) | (0.027) | (0.022) | (0.023) | (0.023) | (0.024) | (0.021) | (0.024) | (0.028) | (0.021) |
| Subjective Cognitive Status | Subjective Cognitive Status | 0.044 | 0.152 *** | 0.116 ** | 0.041 | 0.190 *** | (0.040) | 0.064 | 0.119 * | 0.184 *** | 0.011 | 0.111 * | 0.071 |
| | | (0.061) | (0.058) | (0.054) | (0.070) | (0.060) | (0.066) | (0.055) | (0.062) | (0.061) | (0.065) | (0.061) | (0.063) |
| Size of Social Network | Size of Social Network | (0.002) | 0.001 | 0.000 | (0.002) | 0.000 | (0.001) | (0.001) | 0.003 | 0.000 | 0.001 | 0.000 | 0.001 |
| | | (0.004) | (0.002) | (0.001) | (0.004) | (0.001) | (0.004) | (0.001) | (0.003) | (0.001) | (0.003) | (0.002) | (0.003) |
| Homogeneous Relationships | Intimacy with Relatives | 0.495 ** | (0.108) | 0.116 | 0.101 | (0.077) | 0.556 *** | 0.080 | 0.277 | 0.086 | 0.230 | (0.014) | 0.313 * |
| | | (0.192) | (0.159) | (0.163) | (0.183) | (0.149) | (0.215) | (0.160) | (0.177) | (0.163) | (0.176) | (0.180) | (0.165) |
| | Intimacy with Neighbors | 0.062 | 0.099 | 0.063 | 0.019 | 0.120 | (0.087) | 0.065 | 0.208 | (0.006) | 0.093 | (0.007) | 0.261 |
| | | (0.168) | (0.182) | (0.155) | (0.192) | (0.150) | (0.210) | (0.159) | (0.186) | (0.187) | (0.159) | (0.179) | (0.167) |
| | Intimacy with Acquaintances | 0.197 | 0.012 | 0.161 | 0.026 | 0.032 | 0.055 | 0.040 | 0.128 | 0.034 | 0.121 | (0.185) | 0.383 * |
| | | (0.202) | (0.254) | (0.220) | (0.221) | (0.179) | (0.319) | (0.214) | (0.216) | (0.244) | (0.190) | (0.227) | (0.217) |

**Table 5.** *Cont.*

| Variable Name | Variable Name | The Older | The Younger | Higher Level of Education | Lower Education Level | Health Status Is Better | Poor Health | More Family Members | Fewer Family Members | Higher Household Income | Lower Household Income | Man | Woman |
|---|---|---|---|---|---|---|---|---|---|---|---|---|---|
| Heterogeneous Relationships | Intimacy with Village Officials | 0.170 | 0.251 * | 0.154 | 0.378 ** | 0.269 ** | 0.230 | 0.372 *** | (0.036) | 0.325 ** | 0.124 | 0.388 ** | 0.246 |
| | | (0.153) | (0.144) | (0.139) | (0.171) | (0.133) | (0.185) | (0.138) | (0.172) | (0.153) | (0.147) | (0.156) | (0.156) |
| | Intimacy with Government Personnel | 0.297 | 0.200 | 0.192 | 0.521 ** | 0.193 | 0.670 *** | 0.186 | 0.391 ** | 0.347 * | 0.297 | 0.492 *** | 0.118 |
| | | (0.188) | (0.179) | (0.150) | (0.242) | (0.156) | (0.255) | (0.186) | (0.184) | (0.182) | (0.182) | (0.185) | (0.177) |
| | Intimacy with Social Groups | 1.366 *** | 1.566 *** | 1.454 *** | 1.139 ** | 1.382 *** | 1.049 * | 1.473 *** | 0.926 * | 1.694 *** | 0.944 ** | 0.874 * | 1.940 *** |
| | | (0.482) | (0.571) | (0.479) | (0.530) | (0.459) | (0.571) | (0.462) | (0.508) | (0.564) | (0.451) | (0.447) | (0.570) |
| Constant Term | | (3.910) | −6.907 ** | −7.537 *** | (0.571) | −8.564 *** | (0.851) | −6.935 *** | (3.343) | −9.057 *** | (2.087) | (4.335) | −7.569 *** |
| | | (3.445) | (2.726) | (2.444) | (3.257) | (2.852) | (3.197) | (2.609) | (2.875) | (2.814) | (2.764) | (2.712) | (2.879) |
| Number of Samples | | 163 | 183 | 218 | 128 | 235 | 111 | 204 | 142 | 193 | 153 | 176 | 168 |

Note: ***, ** and * indicate 1%, 5% and 10% significance levels, respectively. Standard errors are in parentheses. All figures are rounded.

### 4.4. Empirical Results

Social networks. (1) Size of social network. "The number of cell phone or WeChat contacts" had little effect on the farmers' willingness to participate in environmental governance. It enhances farmers' willingness to participate in environmental governance. The higher the number of contacts and the larger the size of the social network, the higher the farmers' willingness to participate in environmental governance is likely to be. The scale of the social network affects the speed and breadth of information acquisition, thus, promoting farmers to keep up with the times, enhancing environmental awareness and increasing their willingness to participate in environmental governance. (2) Homogeneous network. The regression results show that the effect of homogeneous relationships on farmers' willingness to participate in environmental governance is not significant, except for "closeness to neighbors" in model 2. However, all of them showed positive effects. This is consistent with the findings of Brunn, S.D. and Li, Y. and B. [24,25,30]. Homogenous relationships exist mainly among families, friends and neighbors who have emotional ties. Their values are convergent and their social circles are highly overlapping. There is information redundancy among people with many interactions or close relationships, and the trust, reciprocity and reputation generated by long-term interactions gradually form an "institutionalized" sediment. This precipitation subjects each other's words and actions to common norms and deepens their adherence to their perceptions. Thus, homogeneous relationships have no significant effect on the farmers' willingness to participate in environmental governance. (3) Heterogeneous relationships. In both Model 1 and Model 2, the heterogeneous relationship passed the significance test. They all have a positive effect on the farmers' willingness to participate in environmental governance. This indicates that the closer the relationship with village cadres, government personnel and social groups, the stronger the willingness of the farmers to participate in environmental governance. A full range of relationships allows the farmers to have access to more information resources and information channels outside the circle of interaction. The more contacts they have with village cadres, government officials or social groups, the more information they learn about environmental governance. The more open their horizons, the higher their level of knowledge about the environment and, therefore, the stronger their willingness to participate in environmental management. However, the results showed that closeness to social groups had a more significant effect on the farmers' willingness to participate in environmental governance than closeness to village cadres and government personnel. Although all three are heterogeneous relationships, farmers may have a deeper sense of class identity, less distance from social groups and greater receptivity to information resources disseminated by social groups than village cadres and government personnel. Thus, the degree of influence of closeness to social groups on the farmers' willingness to participate in environmental governance is much greater than that of village cadres and government personnel. This suggests that the impact of heterogeneous relationships can vary greatly depending on the group with which farmers interact.

Individual characteristics. (1) Age. Age in model 2 passed the significance test at the 10% statistical level and was negatively associated with the farmers' willingness to participate. The older the age, the less receptive and relatively less aware of environmental protection and the less capable of behavior they become. Therefore, it indicates that the older the age, the lower the willingness to participate in environmental governance. (2) Education level. Education level in model 1 passed the significance test at the 5% statistical level and was positively correlated with the willingness to participate in environmental governance. Due to the difference in literacy level, the degree of mastery of environmental knowledge and environmental protection concepts may also vary. The higher the literacy level, the more receptive and more knowledgeable about the environment and, therefore, the stronger the willingness to participate in rural environmental management. (3) Health status. Health status in model 2 passed the significance test at the 10% statistical level and was positively associated with the farmers' willingness to participate in environmen-

tal management. The healthier body is more inclined to participate in environmental governance.

Family characteristics. The number of family members in both Model 1 and Model 2 passed the significance test at the 5% statistical level and had a positive effect on the farmers' willingness to participate. First, the larger the family size, the more social resources they may have access to and the more channels they have to obtain information, which can improve their cognitive level. Second, the larger the household size, the lighter the environmental management tasks that each individual needs to undertake on average, and these tasks are no longer borne by individual farmers. Therefore, the larger the number of family members, the stronger their willingness to govern the environment.

Subjective cognitive status. Subjective cognitive status in model 1 passed the significance test at the 5% statistical level and had a positive effect on the farmers' willingness to participate in environmental governance. The better the subjective cognitive status, the more environmental knowledge they have, the better they can understand the meaning of environmental protection and the benefits of environmental protection. Therefore, people with more environmental knowledge are more willing to participate in rural environmental management.

Subsample regression results analysis. The regression of the subsamples shows that the heterogeneity relationship is relatively stronger for the younger subgroup, the less educated subgroup, the higher household income subgroup and the male subgroup, which passed the significance test at different statistical levels, respectively, and are positively related to the farmers' willingness to participate in environmental governance. The possible reason for this is that younger age means the stronger ability to accept new things as well as cognitive ability, the stronger willingness to participate in environmental governance and also the relatively stronger willingness to socialize externally and, thus, form stronger heterogeneous relationships. The heterogeneous relationship was stronger in the less educated subgroup, probably because the less educated farmers received frequent visits and knowledge dissemination from government staff and village cadres, which increased their contact with each other. The heterogeneous relationship is stronger in the subgroup with higher household income because higher household income indicates that they are also richer in social capital and have more connections with village cadres, government workers and social groups because of the stronger heterogeneous relationship. Furthermore, due to the rich social capital, these people have more advanced concepts and environmental awareness and are more inclined to participate in environmental governance. Compared to women, man subgroups have stronger heterogeneous relationships. This may be due to the traditional concept that males are more inclined to deal with household external affairs and females are more inclined to deal with internal household affairs, hence, the stronger heterogeneous relationships among males.

## 5. Conclusions and Discussion

This paper is based on a field survey of 346 samples from 10 localities in Fujian Province, Anhui Province and Shanxi Province. The influencing factors on farmers' participation in rural environmental governance were studied. The following conclusions were drawn after the analysis.

First, the study results show that 67.6% of farmers are willing to participate in rural environmental governance, and only 32.4% are not willing to participate in rural environmental governance. The farmers' willingness to participate in environmental governance is generally high.

Second, heterogeneous relationships in social networks had a significant effect on the farmers' willingness to participate, while homogeneous relationships had a smaller effect, and heterogeneous relationships had a greater degree of influence than homogeneous relationships. This finding is consistent with the findings of Granovetter and Yubei Li [19,24,25] but contradicts the findings of Yanjie Bian and Hengtong Shi. Heterogeneous relationships can lead to the flow of information resources within different communicative circles, which

can expand the access to information [21,30]. Homogeneous relationships, on the other hand, are communicative interactions within the same communicative circle and tend to only aggravate people's inherent perceptions.

Third, the findings suggest that the strength of the role of social network relationships varies considerably by the interacting group in both homogeneous and heterogeneous relationships. This finding was addressed in the other literature. The effect of closeness to relatives and neighbors on the farmers' willingness to participate in environmental governance was significantly greater than that of closeness to acquaintances; among heterogeneous relationships, the effect of closeness to social groups on the farmers' willingness to participate in environmental governance was much greater than that of closeness to village cadres and government personnel. Based on the results of the study, it can be hypothesized that the strength of ties varies even among relationships with different attributes. Social network relationships with different attributes determine whether they have a significant effect on the farmers' willingness to participate, while the strength of the relationship affects the magnitude of the farmers' willingness to participate.

Fourth, social network size has a positive effect on the farmers' willingness to participate in environmental governance. This finding is consistent with the majority of the literature [24,25,31]. In today's information-explosive Internet era, social network size means more effective information resources are available. The timeliness and extensiveness of farmers' access to information enhances their quality and increases their awareness of their autonomous environment.

Fifth, in addition to the social networks, the farmers' willingness to participate in environmental governance is influenced by age, education level, number of family members and subjective cognitive status. The farmers' personal and family characteristics also have an important impact on their willingness to participate in environmental governance.

At present, the circular economy is more emphasized in industry and less in agriculture, establishing a sense of resource concern, the scientific use of limited resources, and the development of the agricultural circular economy. Protecting the agricultural ecological environment is not only a matter of long-term development for agriculture but also a strategic issue for overall socio-economic development. The government should formulate a practical medium- and long-term development plan for the agricultural recycling economy from the perspective of long-term development. Through the formulation and introduction of relevant policies and regulations, it should strengthen the standard guidance of circular economy development planning. Through in-depth research and analysis, the responsibilities and obligations of the public, enterprises, and the government in the development of the agricultural circular economy should be determined.

First, the government should actively create objective conditions to nurture the farmers' heterogeneous relationships. This is because heterogeneous relationships have a significant impact on promoting the farmers' willingness to enhance environmental management. The serendipity-mindsponge-3D knowledge management theory framework views the human mind as an "information-gathering-processor" [32]. The brain uses information stored in the brain and absorbed from the environment as input and generates items that drive human cognitive processes and behaviors, such as value systems, perceptions, thoughts, feelings, behaviors, etc. [33]. In order to generate creative ideas, the input information needs to go through a multiple filtering system of information in the mind, where information will be evaluated, connected, compared and used as material for the imagination to produce information that is different from the original information, with enough new useful knowledge, wisdom and abilities from the external environment (including other people), which is more likely to increase the probability of obtaining creative ideas [34,35]. Therefore, the farmers are more likely to acquire knowledge and information from heterogeneous relationships in the external environment to make them better at domestic waste disposal. Village cadres should strengthen communication with villagers, actively promote environmental management policies and enhance the dissemination of environmental information. At the same time, relevant government departments should strengthen

internal rural institutions, actively publicize environmental governance policies, strengthen communication with grassroots people and drive grassroots people to participate in rural environmental governance. In addition, voluntary organizations related to environmental governance should be cultivated within rural areas, and voluntary groups belonging to farmers themselves should be established.

Second, the government should introduce relevant incentive policies. If the policy meets more of the core values of the target audience, it will have a better chance of success. The mindspongeconomics assumes that individuals, households, businesses, governments and other actors ultimately make decisions based on their core values and the information they receive, and that the speed of the decision-making process is related to the clarity, detail and priority of the core values in their perceptions. Typical core values include: costs and benefits, environmental values, social values, trust, etc. [36–38]. Based on these core values, the government can introduce a corresponding point policy for domestic waste disposal where villagers who do well in waste separation or participate in volunteer services can earn points, which can be redeemed for household items such as aged vinegar, rice and soap, as well as for services such as coupons for the use of fitness equipment, family doctors and health checkups, when accumulated to a certain value. In this way, the farmers are promoted to improve their own garbage disposal behavior.

Third, the state should make efforts to improve the development of education in rural areas. Improving the literacy level of farmers is necessary to increase their willingness to participate in environmental governance [39–41]. Improving the literacy level of the rural population as a whole will help to improve the willingness to participate and fundamentally improve the environmental situation in rural areas. At the same time, popularizing knowledge about environmental protection and environmental rights in rural areas can greatly enhance farmers' understanding of the environment. Grassroots governments should make greater efforts to convey the key points of environmental-related policies to farmers so that they can improve their environmental knowledge and awareness of their responsibilities and thus strengthen their willingness to participate in rural environmental governance.

**Author Contributions:** Conceptualization, J.L. and Y.W.; methodology, Y.W. and J.L.; software, J.L., Y.W. and S.H.; validation, Y.W.; formal analysis, S.H. and Y.W.; investigation, S.H., Y.W. and J.L.; resources, Y.W. and S.H.; data curation, S.H., Y.W. and J.L.; writing—original draft preparation, J.L. and Y.W.; writing—review and editing, J.L., S.H. and Y.W., visualization, J.L., Y.W. and S.H.; supervision, J.L.; project administration, S.H.; funding acquisition, S.H. All authors have read and agreed to the published version of the manuscript.

**Funding:** This research was funded by the General Project of The National Social Science Fund of China, grant number 20BSH113, Senwei Huang, and the Annual Key Project of Beijing Social Science Foundation, grant number 21JJA003, Qiang Zhang.

**Institutional Review Board Statement:** Not applicable.

**Informed Consent Statement:** Not applicable.

**Data Availability Statement:** All the data used in this study are public.

**Conflicts of Interest:** The authors declare no conflict of interest.

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
