# Peer review of "Study of Farmers’ Willingness to Participate in Environmental Governance Based on Recycling, Reduction and Resourcing"

_sustainability, doi:10.3390/su151410850_

Round 1
Reviewer 1 Report
Reviewer’ comments:
I have some comments on your paper titled “A Survey of Farmers' Willingness to Participate in Environmental Governance Based on Recycling, Reduction and Resourcing”. Your paper could potentially contribute to the literature on environmental management and development, yet it contains some weak parts.
Specific comments:
1. Your paper lacks a conceptual model/framework, which limits the comprehension of your findings. It is possible of using mindspongeconomics and mindsponge theory to build your conceptual framework. These are new social science theories that investigate human behaviors using values of the mindset, employing information from the living environment as an input for the decision making process/mechanism, using both rational and irrational thinking.
2. Your paper structure appears to be inadequate, so please carefully improve this point in the updated version.
3. The discussion section is weak. What mechanism drives farmers to take part in environmental governance? In light of mindspongenomics, the more the program satisfies farmers’ core values, the more likely they are to participate. In this sense, you should use the important theories/frameworks to support your interpretations of the findings. For example, it is possible to use again the SM3D knowledge management system along with mindspongeconomics, mindsponge theory to provide a more in-depth discussion.
4. Please add/supplement a limitation section in the update version.
5. There are some key references that you may consider using for your revising:
· Vuong, Q. H. et al. (2022). Covid-19 vaccines production and societal immunization under the serendipity-mindsponge-3D knowledge management theory and conceptual framework. Humanities and Social Sciences Communications, 9(1), 1–12. https://doi.org/10.1057/s41599-022-01034-6
· Khuc, Quy Van, Mindspongeconomics (September 12, 2022). VIETKAP Working paper Series No. 2022/9, Available at SSRN: http://dx.doi.org/10.2139/ssrn.4453917
· Vuong, Q.-H. Mindsponge Theory; De Gruyter: Berlin, Germany, 2023.
NA
Author Response
We gratefully appreciate for your valuable comments.
1 Comment : Your paper lacks a conceptual model/framework, which limits the comprehension of your findings. It is possible of using mindspongeconomics and mindsponge theory to build your conceptual framework. These are new social science theories that investigate human behaviors using values of the mindset, employing information from the living environment as an input for the decision making process/mechanism, using both rational and irrational thinking.
1 Reply: We have added relevant sociological and economic theoretical foundations.
2.2Theoretical foundation
Starting from Popkin's rational smallholder perspective, farmers' low levels of income and weak economic capacity make it difficult for them to pay for the costs of environmental behavior, resulting in farmers not participating in environmental governance.In addition, "farmers' attitudes toward environmental responsibility vary according to farm size (both physical and economic)." Larger farms are more focused on eco-efficiency and cleaner production. The core concern of the economic factor is the relationship between the profitability of individual farmers and farm households and the cost of environmental behavior. Factors such as household economic resources, loan support and access to relevant subsidies influence the willingness of farmers to participate in the construction of a new low-carbon rural area. Economic conditions play a fundamental role in social development, and unbalanced economic and social development also acts on education, cognition, skills and other factors, thus leading to an uneven regional distribution of farm household participation levels.
The system as a specific code of conduct regulates the space and form of participation of farmers. In environmental governance, "a sound legal system for environmental protection is an important prerequisite for realizing citizens' right to environmental participation". The lack of a systemic legal system makes farmers vulnerable to environmental violations, and the lack of administrative safeguards and judicial remedies as an extension of the system also leads to violations of farmers' environmental rights, making farmers' ecological participation limited. The system also generates trust capital, and institutional trust is a robust promoter of farmers' participation in environmental governance.
The scale of relational reciprocity network formed by socialized small farmers in the acquaintance society is small, so the insufficient stock of social capital can affect the effectiveness of environmental governance. Social capital has a value-oriented function in enabling environmental behavior, and cognitive social capital can shape farmers' pro-environmental values through value internalization and value sharing.
2 Comment: Your paper structure appears to be inadequate, so please carefully improve this point in the updated version.
2 Reply: The structure has been refined in an updated paper.
3 Comment: The discussion section is weak. What mechanism drives farmers to take part in environmental governance? In light of mindspongenomics, the more the program satisfies farmers’ core values, the more likely they are to participate. In this sense, you should use the important theories/frameworks to support your interpretations of the findings. For example, it is possible to use again the SM3D knowledge management system along with mindspongeconomics, mindsponge theory to provide a more in-depth discussion.
3 Reply: The structure has been refined in an updated paper. We incorporated Model Settings into the third section to optimize article structure
4 Comment:Please add/supplement a limitation section in the update version.
4 Reply: We have Re-specified the model and added a limitation section in the section of “3.3Model setting”. And explained why the model was chosen.
5 Comment: There are some key references that you may consider using for your revising。
5 Reply: We have added more updated references in the text.
Reviewer 2 Report
The study addresses the topical issue of the establishing a green low-carbon agricultural economic system based on the principles of Recycling, Reduction and Resourcing. The study provides an interesting insight on a comprehensive green transformation of economic development and achieving carbon neutrality in agricultural. Text is well written and provides an interesting perspective on the issue for potential discussions in the field.
However, I would like to make some remarks: The title of the article should be changed. It should reflect the aim of the survey (it should not contain "Survey "). The theoretical grounding of the issue is insufficient and needs to be supplemented and expanded. See e.g.: ○ Demkova, M., et al. 2022. Potential for Sustainable Development of Rural Communities by Community-Based Ecotourism. A Case Study of Rural Village Pastanga, Sikkim Himalaya, India. GeoJournal of Tourism and Geosites, 43(3), pp. 964-975, https://doi.org/10.30892/gtg.43316-910; ○ Acharya, A. et al. 2021. Assessing the Economic Impact of Tourism and Verdict Ecotourism Potential of the Coastal Belt of Purba Medinipur District, West Bengal. Folia Geographica 63(2), pp. 82-107;; ○ Nurković, R., Rewucki, J. 2018. Economic relations of Bosnia and Herzegovina and Islamic Countries. Folia Geographica 60(2), pp. 49-61;; ○ Lechowski, Ł. 2021. The socio-economic transformations of municipalities in lodz metropolitan area in the context of the construction of motorways and expressways. Folia Geographica 63(1), pp. 40-63;; ○ Lochman, J., Vágner, J. 2022. The Impact on Environmental Sustainability of Catering Facilities for Tourists. Folia Geographica, 64(1), pp. 5-26;; ○ Brunn, S. et.al. 2018. Policy implications of the vagaries in population estimates on the accuracy of sociographical mapping of contemporary Slovak Roma communities. GEOJOURNAL 83 (4) , pp.853-869-
In conclusion, it is a quality study based on a well-developed and original methodology for obtaining and processing relevant data. The paper has a logical structure, relies on relevant sources and provides an interesting perspective on the discussed problem the achieving carbon neutrality in agricultural.
However, the above comments in no way diminish the quality of the study. It is balanced in content, uses correct methods and I definitely recommend it for publication after minor changes.
Author Response
We gratefully appreciate for your valuable comments.
1 Comment : However, I would like to make some remarks: The title of the article should be changed. It should reflect the aim of the survey (it should not contain "Survey "). The theoretical grounding of the issue is insufficient and needs to be supplemented and expanded.
1 Reply: We have added relevant sociological and economic theoretical foundations. And we have added more updated references in the text.
2.2Theoretical foundation
Starting from Popkin's rational smallholder perspective, farmers' low levels of income and weak economic capacity make it difficult for them to pay for the costs of environmental behavior, resulting in farmers not participating in environmental governance.In addition, "farmers' attitudes toward environmental responsibility vary according to farm size (both physical and economic)." Larger farms are more focused on eco-efficiency and cleaner production. The core concern of the economic factor is the relationship between the profitability of individual farmers and farm households and the cost of environmental behavior. Factors such as household economic resources, loan support and access to relevant subsidies influence the willingness of farmers to participate in the construction of a new low-carbon rural area. Economic conditions play a fundamental role in social development, and unbalanced economic and social development also acts on education, cognition, skills and other factors, thus leading to an uneven regional distribution of farm household participation levels.
The system as a specific code of conduct regulates the space and form of participation of farmers. In environmental governance, "a sound legal system for environmental protection is an important prerequisite for realizing citizens' right to environmental participation". The lack of a systemic legal system makes farmers vulnerable to environmental violations, and the lack of administrative safeguards and judicial remedies as an extension of the system also leads to violations of farmers' environmental rights, making farmers' ecological participation limited. The system also generates trust capital, and institutional trust is a robust promoter of farmers' participation in environmental governance.
The scale of relational reciprocity network formed by socialized small farmers in the acquaintance society is small, so the insufficient stock of social capital can affect the effectiveness of environmental governance. Social capital has a value-oriented function in enabling environmental behavior, and cognitive social capital can shape farmers' pro-environmental values through value internalization and value sharing.
Reviewer 3 Report
please see the attachment

must be reviewed by a professional, native proofreader regarding the grammar and the style.
Reviewer 4 Report
1) There are 3 models estimated but what are these models in not properly and clearly stated in the text. Maybe an additional table is recommended.
2) Models are estimated on the whole sample. Maybe it would be prefered to check robustness of the models by applying some sample splitting and ROC curve.
3) Indeed it is not known how the estimated models "fits" the data.
4) A bit more references should be added.
Author Response
We gratefully appreciate for your valuable comments.
1 Comment : There are 3 models estimated but what are these models in not properly and clearly stated in the text. Maybe an additional table is recommended.
1 Reply: The article re-estimates them by means of the logit model and the probit model, and the use of these two models is clearly explained in detail at the beginning of subsection 3.3.
The article used Stata 13.0 to perform logistic stepwise regression on 346 sample data, and in order to test the regression stability, the article was tested by replacing the Probit model. Model 1 is the baseline model and model 2 is the Oprobit model. See Table 4.
Table 4 Regression results
|
Variable Nam |
Variable Name |
Model 1 coefficients |
Model 2 coefficients |
|
Individual Characteristics |
Age |
-0.020 |
-0.010* |
|
|
(0.013) |
(0.005) |
|
|
Education Level |
0.090** |
0.016 |
|
|
|
(0.038) |
(0.017) |
|
|
Health Status |
0.156 |
0.110* |
|
|
|
(0.148) |
(0.064) |
|
|
Family Characteristics |
Number of Family Members |
0.149** |
0.070** |
|
|
(0.072) |
(0.028) |
|
|
Annual HouseholdIncome |
0.017 |
(0.005) |
|
|
|
(0.026) |
(0.011) |
|
|
Village Features |
Is a Project Village |
(0.391) |
0.010 |
|
|
(0.278) |
(0.121) |
|
|
Distance from Town |
0.000 |
(0.006) |
|
|
|
(0.015) |
(0.007) |
|
|
Subjective cognitive status |
Subjective Cognitive Status |
0.106** |
0.021 |
|
|
(0.042) |
(0.016) |
|
|
Size of Social Network |
Size of Social Network |
0.000 |
0.001 |
|
|
|
(0.001) |
0.000 |
|
Homogeneous relationships |
Intimacy with Relatives |
0.108 |
0.077 |
|
|
(0.117) |
(0.051) |
|
|
Intimacy with Neighbors |
0.108 |
0.107** |
|
|
|
(0.117) |
(0.052) |
|
|
Intimacy with Acquaintances |
0.040 |
0.036 |
|
|
|
(0.145) |
(0.060) |
|
|
Heterogeneous relationships |
Intimacy with Village Officials |
0.270** |
0.144*** |
|
|
(0.106) |
(0.047) |
|
|
Intimacy with Government Personnel |
0.319** |
0.106** |
|
|
|
(0.125) |
(0.049) |
|
|
Intimacy with Social Groups |
1.123*** |
0.535*** |
|
|
|
(0.413) |
(0.165) |
|
|
Constant Term |
-6.075*** |
/ |
|
|
(1.847) |
/ |
||
|
Regional Variables |
Control |
Control |
|
|
Number of Samples |
346 |
346 |
|
|
Pseudo R2 |
0.194 |
0.097 |
|
|
Note:***, **, and * indicate 1%, 5%, and 10% significance levels, respectively. Standard errors are in parentheses. All figures are rounded |
|||
2 Comment: Models are estimated on the whole sample. Maybe it would be prefered to check robustness of the models by applying some sample splitting and ROC curve.
2 Reply: Some subsample regressions are added in the fourth part to test the stability. We divided the samples into high and low groups according to the median levels of age, education level, health status, number of household members, household income status, and gender, respectively. See as 4.3 Subsample Regression.
To further test the robustness of the regression results, the article divided the samples into high and low groups according to the median levels of age, education level, health status, number of household members, household income status, and gender, respectively. See Table 5.
Table 5. Subsample regression results
|
Variable Nam |
Variable Name |
The Older |
The Younger |
Higher Level of Education |
Lower Education Level |
Health Status is Better |
Poor Health |
More Family Members |
Fewer Family Members |
Higher Household Income |
Lower Household Income |
Man |
Woman |
|
Individual Characteristics |
Age |
(0.031) |
(0.017) |
(0.005) |
-0.061** |
(0.025) |
(0.012) |
(0.005) |
-0.037* |
(0.001) |
(0.034) |
(0.021) |
(0.018) |
|
|
(0.035) |
(0.025) |
(0.015) |
(0.026) |
(0.016) |
(0.026) |
(0.018) |
(0.020) |
(0.019) |
(0.022) |
(0.017) |
(0.021) |
|
|
Education Level |
0.145** |
0.065 |
0.113 |
0.107 |
0.118** |
0.080 |
0.139** |
0.077 |
0.113* |
0.096* |
0.096* |
0.106* |
|
|
|
(0.059) |
(0.060) |
(0.082) |
(0.107) |
(0.048) |
(0.079) |
(0.058) |
(0.058) |
(0.059) |
(0.057) |
(0.058) |
(0.059) |
|
|
Health Status |
(0.006) |
0.237 |
0.327 |
(0.207) |
0.341 |
(0.400) |
0.319* |
(0.415) |
0.105 |
0.042 |
(0.135) |
0.313 |
|
|
|
(0.212) |
(0.225) |
(0.200) |
(0.234) |
(0.416) |
(0.468) |
(0.191) |
(0.270) |
(0.217) |
(0.212) |
(0.219) |
(0.226) |
|
|
Family Characteristics |
Number of Family Members |
0.084 |
0.174 |
0.138 |
0.211* |
0.215** |
0.037 |
0.293* |
0.104 |
0.164 |
0.155 |
0.228** |
0.083 |
|
|
(0.090) |
(0.131) |
(0.097) |
(0.123) |
(0.098) |
(0.121) |
(0.151) |
(0.279) |
(0.125) |
(0.096) |
(0.111) |
(0.107) |
|
|
Annual HouseholdIncome |
0.106* |
(0.012) |
0.005 |
0.038 |
0.012 |
0.028 |
0.006 |
0.013 |
0.032 |
0.193 |
0.016 |
0.005 |
|
|
|
(0.062) |
(0.030) |
(0.038) |
(0.045) |
(0.028) |
(0.073) |
(0.039) |
(0.045) |
(0.034) |
(0.176) |
(0.043) |
(0.039) |
|
|
Village Features |
Is a Project Village |
(0.661) |
(0.173) |
(0.179) |
-0.877* |
(0.177) |
(0.728) |
(0.401) |
(0.332) |
(0.632) |
(0.208) |
(0.130) |
(0.476) |
|
|
(0.416) |
(0.406) |
(0.376) |
(0.451) |
(0.363) |
(0.497) |
(0.385) |
(0.453) |
(0.411) |
(0.414) |
(0.430) |
(0.415) |
|
|
Distance from Town |
(0.021) |
0.047 |
0.002 |
(0.014) |
(0.007) |
(0.003) |
(0.014) |
0.006 |
0.008 |
(0.023) |
(0.015) |
0.004 |
|
|
|
(0.020) |
(0.037) |
(0.019) |
(0.027) |
(0.022) |
(0.023) |
(0.023) |
(0.024) |
(0.021) |
(0.024) |
(0.028) |
(0.021) |
|
|
Subjective cognitive status |
Subjective Cognitive Status |
0.044 |
0.152*** |
0.116** |
0.041 |
0.190*** |
(0.040) |
0.064 |
0.119* |
0.184*** |
0.011 |
0.111* |
0.071 |
|
|
(0.061) |
(0.058) |
(0.054) |
(0.070) |
(0.060) |
(0.066) |
(0.055) |
(0.062) |
(0.061) |
(0.065) |
(0.061) |
(0.063) |
|
|
Size of Social Network |
Size of Social Network |
(0.002) |
0.001 |
0.000 |
(0.002) |
0.000 |
(0.001) |
(0.001) |
0.003 |
0.000 |
0.001 |
0.000 |
0.001 |
|
|
|
(0.004) |
(0.002) |
(0.001) |
(0.004) |
(0.001) |
(0.004) |
(0.001) |
(0.003) |
(0.001) |
(0.003) |
(0.002) |
(0.003) |
|
Homogeneous relationships |
Intimacy with Relatives |
0.495** |
(0.108) |
0.116 |
0.101 |
(0.077) |
0.556*** |
0.080 |
0.277 |
0.086 |
0.230 |
(0.014) |
0.313* |
|
|
(0.192) |
(0.159) |
(0.163) |
(0.183) |
(0.149) |
(0.215) |
(0.160) |
(0.177) |
(0.163) |
(0.176) |
(0.180) |
(0.165) |
|
|
Intimacy with Neighbors |
0.062 |
0.099 |
0.063 |
0.019 |
0.120 |
(0.087) |
0.065 |
0.208 |
(0.006) |
0.093 |
(0.007) |
0.261 |
|
|
|
(0.168) |
(0.182) |
(0.155) |
(0.192) |
(0.150) |
(0.210) |
(0.159) |
(0.186) |
(0.187) |
(0.159) |
(0.179) |
(0.167) |
|
|
Intimacy with Acquaintances |
0.197 |
0.012 |
0.161 |
0.026 |
0.032 |
0.055 |
0.040 |
0.128 |
0.034 |
0.121 |
(0.185) |
0.383* |
|
|
|
(0.202) |
(0.254) |
(0.220) |
(0.221) |
(0.179) |
(0.319) |
(0.214) |
(0.216) |
(0.244) |
(0.190) |
(0.227) |
(0.217) |
|
|
Heterogeneous relationships |
Intimacy with Village Officials |
0.170 |
0.251* |
0.154 |
0.378** |
0.269** |
0.230 |
0.372*** |
(0.036) |
0.325** |
0.124 |
0.388** |
0.246 |
|
|
(0.153) |
(0.144) |
(0.139) |
(0.171) |
(0.133) |
(0.185) |
(0.138) |
(0.172) |
(0.153) |
(0.147) |
(0.156) |
(0.156) |
|
|
Intimacy with Government Personnel |
0.297 |
0.200 |
0.192 |
0.521** |
0.193 |
0.670*** |
0.186 |
0.391** |
0.347* |
0.297 |
0.492*** |
0.118 |
|
|
|
(0.188) |
(0.179) |
(0.150) |
(0.242) |
(0.156) |
(0.255) |
(0.186) |
(0.184) |
(0.182) |
(0.182) |
(0.185) |
(0.177) |
|
|
Intimacy with Social Groups |
1.366*** |
1.566*** |
1.454*** |
1.139** |
1.382*** |
1.049* |
1.473*** |
0.926* |
1.694*** |
0.944** |
0.874* |
1.940*** |
|
|
|
(0.482) |
(0.571) |
(0.479) |
(0.530) |
(0.459) |
(0.571) |
(0.462) |
(0.508) |
(0.564) |
(0.451) |
(0.447) |
(0.570) |
|
|
Constant Term |
(3.910) |
-6.907** |
-7.537*** |
(0.571) |
-8.564*** |
(0.851) |
-6.935*** |
(3.343) |
-9.057*** |
(2.087) |
(4.335) |
-7.569*** |
|
|
(3.445) |
(2.726) |
(2.444) |
(3.257) |
(2.852) |
(3.197) |
(2.609) |
(2.875) |
(2.814) |
(2.764) |
(2.712) |
(2.879) |
||
|
Number of Samples |
163 |
183 |
218 |
128 |
235 |
111 |
204 |
142 |
193 |
153 |
176 |
168 |
|
Note:***, **, and * indicate 1%, 5%, and 10% significance levels, respectively. Standard errors are in parentheses. All figures are rounded.
3 Comment: Indeed it is not known how the estimated models "fits" the data.
3 Reply: Logistic regression models are nonlinear categorical statistical methods designed explicitly for regression analysis of dichotomous dependent variables and analyze the degree of influence of different factors on the dependent variable through regression modeling. The explanatory variables in the article are "0-1" dichotomous variables, so this model is chosen. For a detailed explanation, see 3.3 Model setting.
Logistic regression models are nonlinear categorical statistical methods designed explicitly for regression analysis of dichotomous dependent variables and analyze the degree of influence of different factors on the dependent variable through regression modeling. Therefore, the paper uses a logistic stepwise regression model to estimate the regression parameters using the maximum likelihood estimation method. The regression parameters were estimated using maximum likelihood estimation. According to the requirements of the logistic regression model, suppose … is a set of vectors associated with . Let be the probability of the occurrence of the willingness to participate, and taking the ratio logarithmically to get is the logistic transformation of P, labeled as :
|
|
(1) |
|
|
|
in equation (1) denotes a variable with dichotomous nature. In order to clearly and concisely estimate the influence of social networks on the probability of occurrence of farmers' willingness to participate in environmental governance , the willingness to participate is simplified to a 0-1 type dependent variable, where the variable is 1, representing farmers' willingness to participate, and the variable is 0, representing farmers' unwillingness to participate. a is a constant term, i.e., the natural logarithm of the ratio when the independent variable takes all values of 0. denotes the th factor affecting farmers' willingness to participate. is the partial regression coefficient of logistic regression, which indicates the degree of influence of variable on or .
4 Comment:A bit more references should be added.
4 Reply: We have added more updated references in the text.
Round 2
Reviewer 1 Report
Thank you for addressing some of my comments/suggestions. I think the revised manuscript is improved to some degree. However, it seems that your revision is insufficient. That is, your revision did not include some relevant references suggested, such as mindsponge theory (https://sciendo.com/book/9788367405157), mindspongeconomics (https://papers.ssrn.com/sol3/papers.cfm?abstract_id=4453917), and SM3D knowledge management system (https://www.nature.com/articles/s41599-022-01034-6; https://www.sciencedirect.com/science/article/abs/pii/S1871187123001190), even though your paper topic directly engages in the decision-making process/mechanism using information from the living environment. In this sense, your work would be strengthened if you further elaborate/revise your discussions utilizing those reference as support in the next update.
NA
Author Response
We gratefully appreciate for your valuable comments.
1 Comment :However, it seems that your revision is insufficient. That is, your revision did not include some relevant references suggested, such as mindsponge theory, mindspongeconomics, and SM3D knowledge management system, even though your paper topic directly engages in the decision-making process/mechanism using information from the living environment. In this sense, your work would be strengthened if you further elaborate/revise your discussions utilizing those reference as support in the next update.
1 Reply: We have added to the discussion section the relevant references mentioned in the literature as well as the relevant theories. The details are as follows:
Firstly, the government should actively create objective conditions to nurture farmers' heterogeneous relationships. This is because heterogeneous relationships have a significant impact on promoting farmers' willingness to enhance environmental management. The serendipity-mindsponge-3D knowledge management theory framework views the human mind as an "information-gathering-processor"[32]. The brain uses information stored in the brain and absorbed from the environment as input and generates items that drive human cognitive processes and behaviors, such as val-ue systems, perceptions, thoughts, feelings, behaviors, etc.[33]. In order to generate creative ideas, the input information needs to go through a multiple filtering system of in-formation in the mind, where information will be evaluated, connected, compared, and used as material for the imagination to produce information that is different from the original information, with enough new useful knowledge, wisdom, and abilities from the external environment (including other people) More likely to increase the probability of getting creative ideas[34,35]. Therefore, farmers are more likely to acquire knowledge and information from heterogeneous relationships in the external environment to make them better at domestic waste disposal. Village cadres should strengthen communication with villagers, actively promote environmental management policies, and enhance the dissemination of environmental information. At the same time, relevant government departments should strengthen internal rural institutions, actively publicize environmental governance policies, strengthen communication with grassroots people, and drive grassroots people to participate in rural environmental governance. In addition, voluntary organizations related to environmental governance should be cultivated within rural areas, and voluntary groups belonging to farmers themselves should be established.
Secondly, the government should introduce relevant incentive policies. If the pol-icy meets more of the core values of the target audience, it will have a better chance of success. The mindspongeconomics assumes that individuals, households, businesses, governments, and other actors ultimately make decisions based on their core values and the information they receive, and that the speed of the decision-making process is related to the clarity, detail, and priority of the core values in their perceptions. Typical core values include: costs and benefits, environmental values, social values, trust, etc.[36] . Based on these core values, the government can introduce a corresponding point policy for domestic waste disposal, where villagers who do well in waste separation or participate in volunteer services can earn points, which can be redeemed for household items such as aged vinegar, rice, and soap, as well as for services such as coupons for the use of fitness equipment, family doctors, and health checkups, when accumulated to a certain value. In this way, farmers are promoted to improve their own garbage disposal behavior.
References:
[32]Blau P M. Exchange and power in social life[J]. American Journal of Sociology, 1967.
[33]Sonia S M G. Trust, satisfaction, relational norms, opportunism and dependence as antecedents of employee organizational commitment[J]. Contaduría Y Administración, 2013, 58:11-38.
[34]Herb S, Hartmann E. Opportunism risk in service triads – a social capital perspective[J]. International Journal of Physical Distribution & Logistics Management, 2014, 44(3):242 - 256.
[35] Shi Hengtong,Sui Danchen,Wu Haixia,Zhao Minjuan. The influence of social capital on farmers' participation in watershed ecological management behavior:An example from the Heihe River Basin[J]. China Rural Economy,2018(01):34-45.
[36]Tan YANZHI, Zhang ZHAO. Social networks, informal finance, and multidimensional poverty among farm households[J]. Finance and Economics Research,2017,43(03):43-56.
Reviewer 3 Report
Ok
Author Response
Thank you very much for your comment!
Reviewer 4 Report
The paper was improved, so I can suggest acceptance
Author Response
Thank you very much for your comment
Round 3
Reviewer 1 Report
Thank you for your carefully addressing my comments and the updated version is much improved. I have no further comment on your work.
NA